# Retro-2 protects cells from ricin toxicity by inhibiting ASNA1-mediated ER targeting and insertion of tail-anchored proteins

David W Morgens[1†], Charlene Chan[2†], Andrew J Kane[2], Nicholas R Weir[2], Amy Li[1], Michael M Dubreuil[3], C Kimberly Tsui[1], Gaelen T Hess[1], Adam Lavertu[4], Kyuho Han[1], Nicole Polyakov[2], Jing Zhou[2], Emma L Handy[5], Philip Alabi[5], Amanda Dombroski[5], David Yao[1], Russ B Altman[1,6], Jason K Sello[5]*, Vladimir Denic[2]*, Michael C Bassik[1,3,7]*

[1]Department of Genetics, Stanford University, Stanford, United States; [2]Department of Molecular and Cellular Biology, Northwest Labs, Harvard University, Cambridge, United States; [3]Program in Cancer Biology, Stanford University, Stanford, United States; [4]Biomedical Informatics Training Program, Stanford University, Stanford, United States; [5]Department of Chemistry, Brown University, Providence, United States; [6]Bioengineering, Stanford University, Stanford, United States; [7]Stanford University Chemistry, Engineering, and Medicine for Human Health (ChEM-H), Stanford, United States

*For correspondence:
jason_sello@brown.edu (JKS);
vdenic@mcb.harvard.edu (VD);
bassik@stanford.edu (MCB)

[†]These authors contributed equally to this work

Competing interests: The authors declare that no competing interests exist.

**Abstract** The small molecule Retro-2 prevents ricin toxicity through a poorly-defined mechanism of action (MOA), which involves halting retrograde vesicle transport to the endoplasmic reticulum (ER). CRISPRi genetic interaction analysis revealed Retro-2 activity resembles disruption of the transmembrane domain recognition complex (TRC) pathway, which mediates post-translational ER-targeting and insertion of tail-anchored (TA) proteins, including SNAREs required for retrograde transport. Cell-based and in vitro assays show that Retro-2 blocks delivery of newly-synthesized TA-proteins to the ER-targeting factor ASNA1 (TRC40). An ASNA1 point mutant identified using CRISPR-mediated mutagenesis abolishes both the cytoprotective effect of Retro-2 against ricin and its inhibitory effect on ASNA1-mediated ER-targeting. Together, our work explains how Retro-2 prevents retrograde trafficking of toxins by inhibiting TA-protein targeting, describes a general CRISPR strategy for predicting the MOA of small molecules, and paves the way for drugging the TRC pathway to treat broad classes of viruses known to be inhibited by Retro-2.
DOI: https://doi.org/10.7554/eLife.48434.001

## Introduction

Eukaryotic cells destroy many endocytosed pathogens by targeting them to the lysosome. The plant toxin ricin, however, evades this fate by being sorted into early endosome-derived vesicles destined for fusion with the trans-Golgi network (TGN). Following further retrograde transport from the Golgi to the endoplasmic reticulum (ER), ricin retrotranslocates into the cytosol where it catalytically disables ribosomes, halting translation and killing the cell (*Crispin et al., 2009*; *Bassik et al., 2013*; *Sandvig and van Deurs, 2005*; *Lord et al., 2003*; *Johannes and Popoff, 2008*). A high-throughput screen of small molecules first identified Retro-2 as an inhibitor of ricin and Shiga-like toxins that halts their retrograde progression to the ER by trapping them in the early endosome (*Stechmann et al., 2010*). Remarkably, Retro-2 was able to rescue mice from lethal doses of ricin

and was tolerated at high doses. Subsequent studies of the bioactive cyclic derivative of Retro-2 (hereafter Retro-2) (*Park et al., 2012*; *Nelson et al., 2013*) and its analogs have also revealed the ability of these compounds to interfere with infection of papillomaviruses (such as HPV16) (*Lipovsky et al., 2013*) and polyomaviruses (such as JCPyV, BKPyV, and simian virus 40) (*Nelson et al., 2013*), which are known to rely on retrograde transport during infection.

While the mechanism of action (MOA) of Retro-2 is not well understood, Retro-2 has been shown to affect the localization of Syntaxin 5 (STX5) and less acutely Syntaxin 6 (*Stechmann et al., 2010*). STX5 – a target-SNARE – is required for fusion of retrograde vesicles with the TGN, and knockdown of its expression is sufficient to trap ricin in the early endosome (*Amessou et al., 2007*; *Dascher and Balch, 1996*; *Suga et al., 2005*; *Norlin et al., 2016*; *Bennett et al., 1993*). Most SNAREs like STX5 are tail-anchored (TA) proteins – a class of membrane proteins with a single, C-terminal transmembrane domain (TMD) that are post-translationally targeted to the ER for insertion by the transmembrane domain recognition complex (TRC) pathway (*Hegde and Keenan, 2011*; *Denic, 2012*; *Cho et al., 2018*). Notably, our previous genome-wide CRISPR/Cas9 deletions screens have identified STX5 and its ER targeting factor, ASNA1 (also known as TRC40) among the top hits that confer resistance to ricin (*Morgens et al., 2017*).

A variety of genetic approaches have been used in mammalian cells to successfully define the MOA of novel small molecules with therapeutic potential (*Acosta-Alvear et al., 2015*; *Matheny et al., 2013*; *Sidrauski et al., 2015*; *Jost and Weissman, 2018*; *Deans et al., 2016*; *Pabon et al., 2018*). In yeast, novel small molecules have been studied by measuring the extent to which the small molecule phenotypes are modified by the presence of a defined, single mutation across the yeast genome. The resulting compound's genetic profile is analogous to a gene's genetic profile, obtained by measuring a particular gene's mutant phenotype in the presence of many second mutations. Remarkably, the genetic profile of the compound often resembles the genetic profile of the compound's target, allowing for target identification (*Giaever et al., 1999*; *Parsons et al., 2004*; *Parsons et al., 2006*; *Hillenmeyer et al., 2008*; *Costanzo et al., 2010*; *Hoepfner et al., 2014*; *Lee et al., 2014*; *Wildenhain et al., 2015*; *Simpkins et al., 2018*). Here, we developed a conceptually analogous genetic profiling approach in mammalian cells and applied it to Retro-2. Analysis of single and paired-gene CRISPRi screens revealed a robust link between Retro-2 and TA protein biogenesis mediated by the TRC pathway. Flow cytometry and quantitative cell microscopy showed that both *ASNA1* gene deletion and Retro-2 treatment resulted in destabilization of a fluorescent TA protein reporter and decreased abundance of endogenous STX5 at the Golgi. Targeted mutagenesis of the ASNA1 genomic locus using a dCas9-AID*Δ fusion (CRISPR-X) identified a point mutation that conferred resistance to Retro-2 in both ricin cytoprotection and fluorescent reporter assays. Finally, using biochemical reconstitution approaches, we demonstrated that Retro-2 obstructed TA protein delivery to the ER targeting factor ASNA1 (TRC40), and this activity was blocked by the resistant mutation in ASNA1. Collectively, these findings support a model in which Retro-2 directly inhibits ASNA1, leading to inefficient ER targeting of TRC pathway clients such as STX5, which ultimately prevents retrograde trafficking of ricin and protects the cell.

## Results

### Genetic profiling reveals that Retro-2 treatment resembles TRC pathway inhibition

Previously, potential drug targets have been defined in yeast by looking for correlations between the chemical-genetic profile of a drug and that of its target (*Giaever et al., 1999*; *Parsons et al., 2004*; *Parsons et al., 2006*; *Hillenmeyer et al., 2008*; *Costanzo et al., 2010*; *Hoepfner et al., 2014*; *Lee et al., 2014*; *Wildenhain et al., 2015*; *Simpkins et al., 2018*). To obtain a chemical-genetic profile of Retro-2, we used CRISPRi to measure the effect of Retro-2 on the ricin phenotypes of 288 hits from a previous genome-wide shRNA screen in the human leukemia cell line K562 (*Bassik et al., 2013*). Using established CRISPRi sgRNA designs (*Horlbeck et al., 2016*), we created a lentiviral library comprising 10 sgRNAs per gene along with 2000 non-targeting and safe-targeting controls (*Morgens et al., 2017*), which we installed into K562 cells engineered to express dCas9-KRAB (*Gilbert et al., 2014*). We then grew infected K562 cells in replicate in the presence of Retro-2 or in the presence of both Retro-2 and ricin (*Figure 1A*). Additional untreated and ricin-only

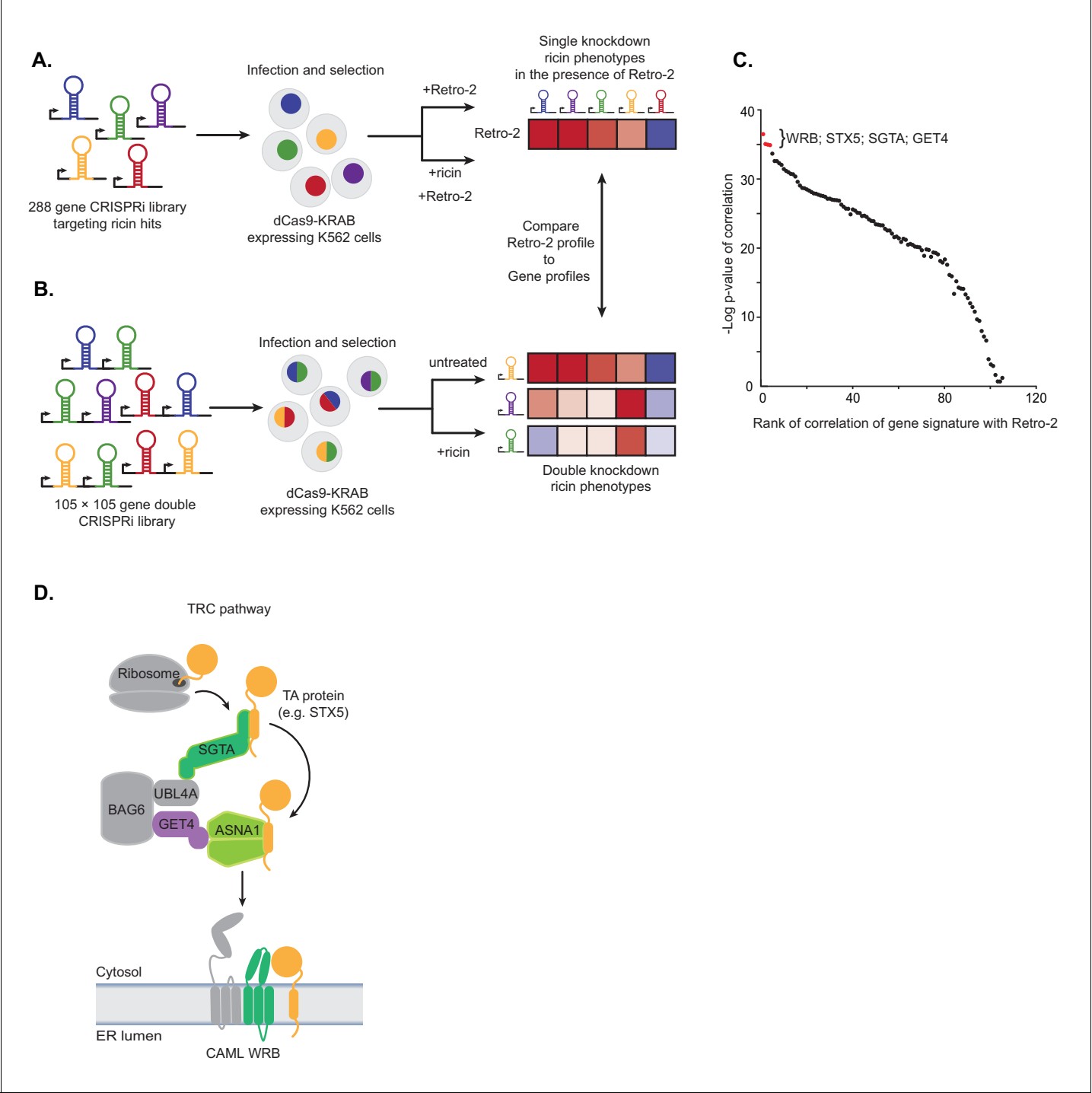

**Figure 1.** Single and paired-gene CRISPRi screens implicate TRC pathway inhibition as the MOA of Retro-2. (**A**) Schematic of single-gene CRISPRi screen. A 288 gene library with 10 guides per gene targeting previously identified ricin hits and 2000 negative controls was lentivirally infected into a K562 cell line expressing a dCas9-KRAB fusion. The pool was then grown in replicate in the presence of 10 μM Retro-2 and presence or absence of 2.5 ng/μL ricin. The ricin phenotypes of the gene knockdowns in the presence of Retro-2 yielded a genetic profile of Retro-2. (**B**) Schematic of paired-gene CRISPRi screen. A library of 105 × 105 genes with three guides per gene and 50 negative controls were lentivirally infected into a K562 cell line expressing a dCas9-KRAB fusion. The pool was then grown in replicate in the presence or absence of ricin. For each of the genes included, the ricin phenotype of the double knockdowns represent a genetic profile. (**C**) Summary of paired-guide screen results. The genetic profile of each gene in the paired-gene CRISPRi screen (the ricin phenotype of each other gene in that background) was correlated (Pearson) with the genetic profile of Retro-2 (the ricin phenotype of each gene in the presence of Retro-2 as measured in the single-gene CRISPRi screen). The x-axis is the rank of the Pearson

*Figure 1 continued on next page*

*Figure 1 continued*

correlation coefficient. The y-axis is the negative log10 p-value of the correlation. The top four ranked genes are labeled and highlighted in red. (D) Schematic of the TRC pathway with the candidates from the single and pair-gene CRISPRi screens highlighted in color.

DOI: https://doi.org/10.7554/eLife.48434.002

The following source data and figure supplement are available for figure 1:

**Source data 1.** Sequencing counts for single gene screens.

DOI: https://doi.org/10.7554/eLife.48434.004

**Source data 2.** Results for single gene screens.

DOI: https://doi.org/10.7554/eLife.48434.005

**Source data 3.** Sequencing counts for double gene screens.

DOI: https://doi.org/10.7554/eLife.48434.006

**Source data 4.** Results for double gene screens.

DOI: https://doi.org/10.7554/eLife.48434.007

**Figure supplement 1.** Genetic profile analysis of ricin phenotypes for Retro-2 and candidate genes.

DOI: https://doi.org/10.7554/eLife.48434.003

replicates were included as controls. Using a maximum likelihood estimator (casTLE; see Materials and methods), we compared the enrichment of sgRNAs between conditions (*Morgens et al., 2016*), measuring the ricin phenotype of each gene knockdown in the absence and presence of Retro-2, as well as the effect of the knockdown on the activity of Retro-2 (*Figure 1—figure supplement 1A–C*; *Figure 1—source datas 1* and *2*). The ricin phenotypes of 288 gene knockdowns in the presence of Retro-2 yielded a genetic profile of Retro-2, which we compared to profiles of candidate genes, as described below.

To measure genetic profiles for candidate Retro-2 target genes, we performed a paired-gene CRISPRi screen (*Figure 1B*). Using our previously established paired-guide platform (*Han et al., 2017*), we constructed a library containing CRISPRi guides targeting 105 × 105 pairs of genes with three guides per gene. We chose 100 of these genes based on having the strongest effect on the activity of Retro-2 (*Figure 1—figure supplement 1C*). Since ASNA1 was one of the top genes in this unbiased group, we also included an additional five genes comprising four additional TRC pathway components (GET4, WRB, CAMLG, and SGTA) and STX5, a TRC pathway substrate that has been shown to relocalize following Retro-2 treatment (*Stechmann et al., 2010*). Following installation of this library into K562 cells expressing dCas9-KRAB (*Gilbert et al., 2014*), we grew the pool of infected cells in replicate in the presence or absence of ricin and monitored the phenotype of the double knockdowns by comparing the enrichment of paired guides between conditions using casTLE (*Morgens et al., 2016*). For each of the 105 genes included, these double phenotypes represent a genetic profile that we compared to the genetic profile of Retro-2 measured in the single-guide CRISPRi screen (*Figure 1A,C*; *Figure 1—figure supplement 1B,C*); this allowed us to identify genes whose knockdown most closely resemble Retro-2 treatment.

The genes whose genetic profile correlated most strongly with the profile of Retro-2 were WRB, STX5, SGTA, and GET4 (*Figure 1C*; *Figure 1—figure supplement 1D–F*; *Figure 1—source datas 3* and *4*). Notably, the genetic profiles of ASNA1 and CAMLG were uninformative (and did not correlate with Retro-2) (*Figure 1—figure supplement 1G*); this is likely because the guides targeting these genes individually had extremely protective ricin phenotypes (independent of the second guide), suggesting either that their phenotypes cannot be modified or the variance is outside the dynamic range of this experiment. Regardless, our analysis revealed that Retro-2 treatment closely resembles genetic perturbation of the TRC pathway (*Figure 1D*). On the basis of this finding, we hypothesized that Retro-2 protects cells against ricin by inhibiting the TRC pathway, thus interfering with ER targeting and insertion of TA proteins, including the SNARE STX5. This would lead to protection against ricin by disrupting the toxin's retrograde transport to the ER. We proceeded to test three predictions of this model: 1) Retro-2 treatment should disrupt biogenesis of TRC pathway substrates, 2) mutations in the relevant TRC pathway component should confer resistance to Retro-2, and 3) Retro-2 should disrupt a specific step along the TRC pathway.

## Retro-2 inhibits targeting of newly-synthesized TRC pathway substrates

To test whether Retro-2 disrupts the biogenesis of TRC pathway substrates, we exploited the molecular triage decision that channels newly-synthesized TA proteins that are not efficiently delivered to ASNA1 to degradation (*Shao et al., 2017*). To examine if Retro-2 induces this form of TA protein instability, we adapted an established approach (*Guna et al., 2018*; *Chitwood et al., 2018*) by constructing a doxycycline (dox)-inducible cassette expressing green fluorescent protein (GFP) linked by a self-cleaving P2A peptide to a red fluorescent protein (RFP) fused to the C-terminal SEC61B TMD sequence (GFP-2A-RFP-SEC61B$_{TMD}$) (*Figure 2A*). When expressed in cells, the RFP-SEC61B$_{TMD}$ fusion protein will be inserted into the ER membrane as a TA protein, but failed insertion will result in its degradation and selective loss of RFP signal relative to GFP. We then lentivirally delivered GFP-2A-RFP-SEC61B$_{TMD}$ into wildtype and ASNA1 knockout (*ASNA1$^{KO}$*) HEK293T cells (*Figure 2—figure supplement 1A*). As expected, we observed by flow cytometry significantly lower RFP: GFP ratio in the *ASNA1$^{KO}$* cells following dox treatment, confirming the dependence of the reporter on the TRC pathway (*Figure 2B*). Strikingly, pre-treatment of wild-type cells with either Retro-2 or a hyperactive analog of Retro-2, DHQZ36.1 (*Craig et al., 2017*), resulted in a comparable RFP:GFP ratio decrease that was not further aggravated by the additional absence of ASNA1 (*Figure 2B*; *Figure 2—figure supplement 1B,C*). In contrast, we detected no significant reporter changes when we treated cells with an inactive analog of Retro-2, DHQZ5 (*Carney et al., 2014*)(*Figure 2—figure supplement 1B*) or when we used a control reporter lacking the TMD (*Figure 2B*).

We next tested the prediction that the sensitivity of the TMD to DHQZ36.1 is not solely determined by its amino acid sequence but additionally relies on its positional context within the protein. Previous in vitro work with SEC61B showed that recognition by ASNA1 can be abolished by appending a C-terminal sequence, leading to preferential TMD engagement by the signal recognition particle (SRP) on the ribosome (*Stefanovic and Hegde, 2007*). To switch the ER targeting specificity of SEC61B TMD in cells by a similar approach, we fused the blue fluorescent protein (BFP) to the C-terminus of GFP-RFP-SEC61B$_{TMD}$ (GFP-2A-RFP-SEC61B$_{TMD}$-BFP) and confirmed that the resulting co-translational reporter was no longer destabilized in *ASNA1$^{KO}$* cells (*Figure 2C*). Consistent with our model, this modification also conferred reporter resistance to DHQZ36.1 in both wildtype and *ASNA1$^{KO}$* cells (*Figure 2D*).

To test our model on the endogenous STX5 protein, we measured STX5 levels at the Golgi upon Retro-2 treatment or genetic perturbation of ASNA1. First, we established HeLa cells expressing either shRNAs targeting ASNA1 or scrambled negative control shRNAs and demonstrated that only the former knockdown conferred resistance to ricin toxicity that was comparable to that caused by Retro-2 (*Figure 3—figure supplement 1A*). Immunofluorescence and confocal microscopy revealed a reduction in STX5 fluorescence density at the Golgi upon Retro-2-treatment of control cells that was comparable to that observed in untreated ASNA1 knockdown cells (*Figure 3A,B*; *Figure 3—figure supplement 1C*). Combining Retro-2 treatment with ASNA1 knockdown resulted in a more severe loss of STX5 at the Golgi (*Figure 3A,B*; *Figure 3—figure supplement 1C*). We also monitored the activity of the hyperactive Retro-2 analog DHQZ36.1 (*Craig et al., 2017*) and observed that it caused a severe loss of Golgi STX5 similar to Retro-2-treated ASNA1 knockdown cells (*Figure 3B*; *Figure 3—figure supplement 1B,C*). Taken together, our fluorescent protein stability reporter and cell microscopy data suggest that Retro-2 and DHQZ36.1 disrupt biogenesis of TRC pathway clients leading to their enhanced degradation and altered subcellular localization, which could explain their effect on STX5 abundance at the Golgi.

## CRISPR-X screen identifies point mutants of ASNA1(TRC40) resistant to Retro-2

To test if the TRC pathway is the functional target of Retro-2 in cells, we attempted to identify mutations in ASNA1 that result in resistance to the compound. We focused on ASNA1 for several reasons. First, ASNA1 is the only TRC pathway component known to interact with small molecules as an ATPase enzyme. Second, we noted that ASNA1 knockdown sensitizes cells to growth inhibition by high concentrations of Retro-2 even in the absence of ricin (*Figure 3—figure supplement 1A*) and that high doses of DHQZ36.1 are toxic to cell growth even in wildtype cells (*Figure 3—figure supplement 1B*). Along with our previous result that ASNA1 is an essential gene for cell growth in CRISPR/Cas9 screens in K562 (*Morgens et al., 2017*), these observations suggested that the toxicity

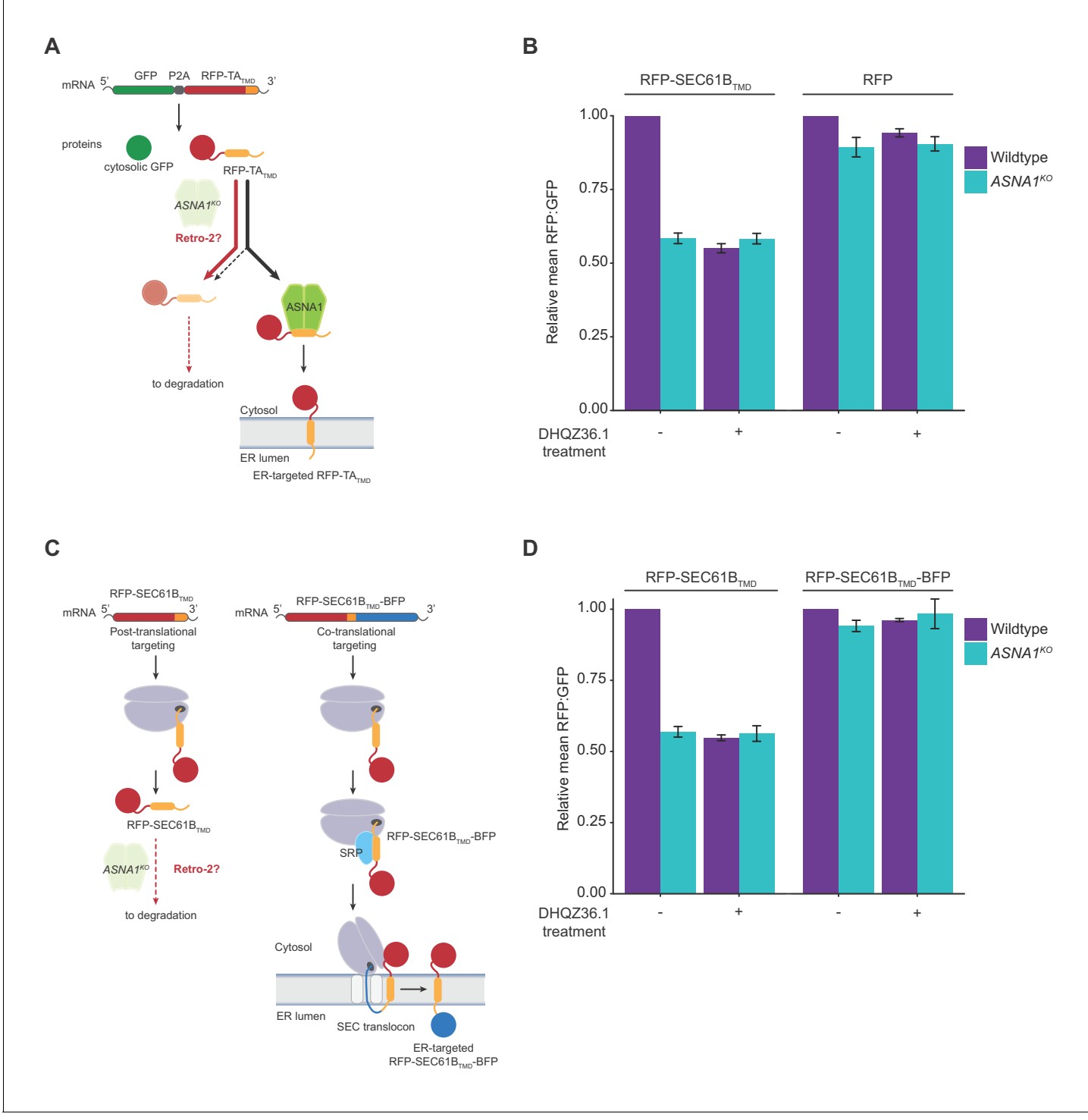

**Figure 2.** Retro-2 diverts newly-synthesized TA proteins from ER targeting to degradation. (**A**) Schematic of the dual-colour reporter consisting of a self-cleaving P2A peptide between a GFP and a RFP with a C-terminal TMD. Genetic and chemical perturbations to targeting pathways will promote destabilization of RFP-TA$_{TMD}$, as TA proteins are diverted for degradation by accessory TRC pathway components (not shown) if they are not efficiently captured by ASNA1. (**B**) Wildtype and *ASNA1$^{KO}$* HEK293T cell lines with indicated reporters were pre-treated with 3 μM DHQZ36.1 for 1 hr prior to induction with dox for approximately 18 hr and FACS analysis. Shown are bar graphs of reporter RFP to GFP ratios with standard deviations derived from three experiments as relative means to their corresponding mock-treated wildtypes. (**C**) Schematic of the post-translational and co-translational ER targeting of RFP-SEC61B$_{TMD}$. and RFP- SEC61B$_{TMD}$-BFP, respectively. (**D**) Cells with indicated genotypes and reporters were treated and analyzed as in part b).

*Figure 2 continued on next page*

*Figure 2 continued*

DOI: https://doi.org/10.7554/eLife.48434.008

The following figure supplement is available for figure 2:

**Figure supplement 1.** ASNA1 knockout and Retro-2/DHQZ36.1 treatment destabilize the transmembrane domain of STX5.

DOI: https://doi.org/10.7554/eLife.48434.009

of Retro-2 and DHQZ36.1 may be mediated by strong inhibition of ASNA1. We used this toxicity to select for ASNA1 alleles resistant to growth inhibition by DHQZ36.1 by first randomly mutagenizing the ASNA1 coding region with CRISPR-directed diversification. Here, a library of sgRNAs tiling the ASNA1 coding region was lentivirally infected into a K562 cell line stably expressing an N-terminal, dCas9-AID*Δ fusion (CRISPR-X) (*Hess et al., 2016*), which results in a high frequency of diverse point mutations at the genomic locus being targeted. We then grew the mutagenized cell population in the presence of toxic doses of DHQZ36.1 for ~4 weeks and deep sequenced the ASNA1 cDNA to identify variant ASNA1 alleles (*Figure 4A*). By this method, we detected a number of distinct ASNA1 mutations which appeared in a moderately high fraction of selected cells relative to the starting

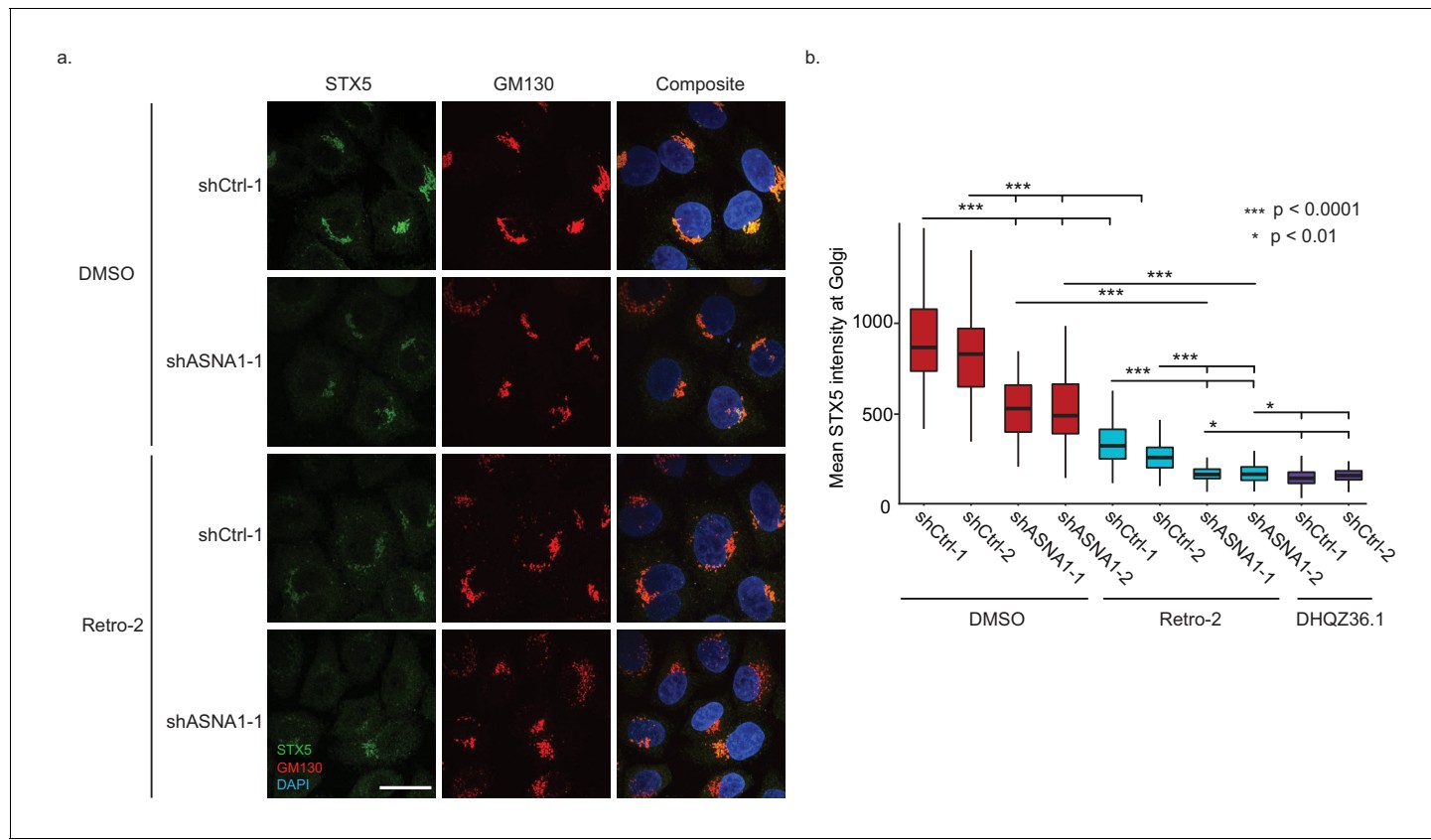

**Figure 3.** ASNA1 knockdown and Retro-2 treatment both decrease the abundance of Golgi-localized STX5, an ASNA1 substrate. (**A**) HeLa cells expressing either ASNA1-targeting or scrambled control (shCtrl) shRNAs were treated for 24 hr with DMSO or 10 µM Retro-2 before fixation and staining for STX5, a Golgi marker (GM130), and a nuclear marker (DAPI). Shown are maximal signal projections of z-stacked confocal micrographs taken with a 100× objective and made without contrast or LUT adjustments. Scale bar represents 25 µm and applies to all images. (**B**) HeLa cells were treated as in part a) but including additional shRNAs and the hyperactive Retro-2 analog DHQZ36.1. Images were collected using a 60× objective and quantitatively analyzed. Shown are box plots of per cell mean STX5 intensity at GM130-marked Golgi for the indicated treatments. Asterisks specify significant differences between treatments as calculated by the MW U test.

DOI: https://doi.org/10.7554/eLife.48434.010

The following figure supplement is available for figure 3:

**Figure supplement 1.** Effect of ASNA1 knockdown or Retro-2 and DHQZ36.1 in HeLa cells.

DOI: https://doi.org/10.7554/eLife.48434.011

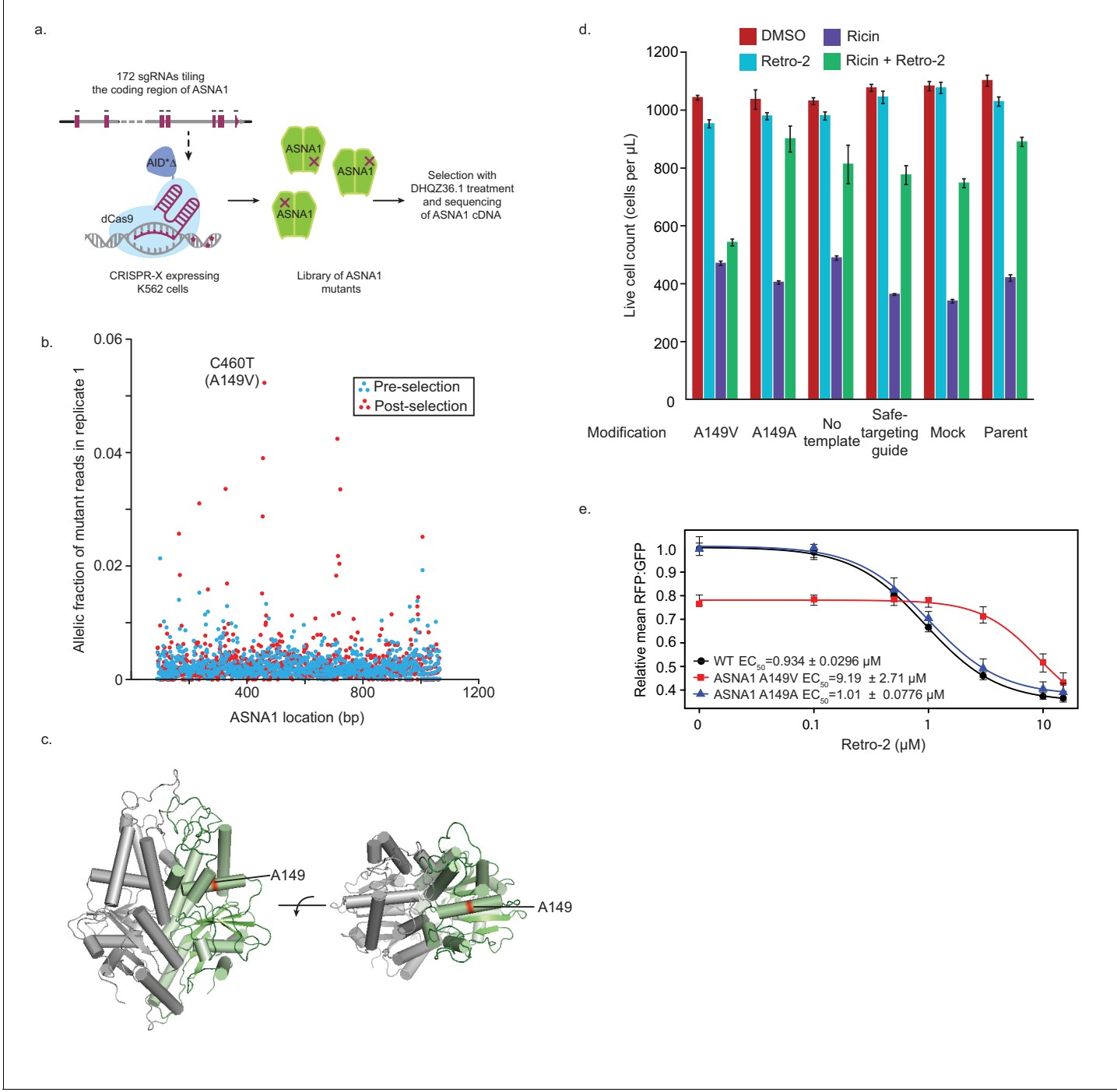

**Figure 4.** Isolation and characterization of A149V ASNA1, a Retro-2-resistance allele. (**A**) Schematic of ASNA1 mutagenesis by CRISPR-X. A 172 sgRNA library tiling the ASNA1 coding region was lentivirally infected in a K562 cell line expressing a dCas9-AID*Δ N-terminal fusion. The pool was then grown in duplicate in the presence of 20 μM DHQZ36.1 for 4 weeks. ASNA1 was amplified separately from cDNA of naive and treated cell populations before being sequenced by Nextera XT. (**B**) Plot of the frequency of ASNA1 alleles across the gene-body (x-axis) in the initial CRISPR-X population (pre-selection) and in one replicate of the selected population. Allelic fraction was calculated by determining the per-base variant frequency that is the number of reads which contain a mutated base at a given position vs the number of reads which contain the wildtype base at that position. Bases which had less than 500× read coverage were excluded, resulting in no resolution of the 5'/3' ends of ASNA1. The top selected mutation in both replicates is highlighted. (**C**) A homology model of human ASNA1 (SWISS-MODEL: O43681) with the location of the top mutated residue, A149, enriched after selection by DHQZ36.1 toxicity highlighted in red on one ASNA1 (colored in green) of the homodimer. (**D**) A149V was installed by homology directed repair in a K562 cell line expressing Cas9. The resulting mutant line was treated with ricin toxin in the presence of 10 μM Retro-2 or DMSO. Live cells were counted using forward/side scatter by cytometry. Shown are bar graphs of the mean with standard error from three technical

*Figure 4 continued on next page*

*Figure 4 continued*

replicates. Also shown are data for five control lines treated in the same way. See Materials and methods for further details. (E) Clonal wildtype (black circles), ASNA1 A149V (C460T) (red squares), and ASNA1 A149A (C461T) (blue triangles) K562 cell lines with the GFP-2A-RFP-SEC61B$_{TMD}$ reporter were pre-treated with Retro-2 with the indicated concentrations for 1 hr prior to induction with dox for approximately 18 hr and FACS analysis. Shown are the dose-response curves for the reporter RFP to GFP ratios as relative means (three experiments) to mock-treated wildtype cells. The dose response was modeled using the four-parameter logistic regression to determine the half maximal effective concentration (EC$_{50}$ ± standard error). Error bars for the means represent the standard error calculated from the four-parameter logistic regression.

DOI: https://doi.org/10.7554/eLife.48434.012

The following source data and figure supplements are available for figure 4:

**Source data 1.** Results for CRISPR-X experiments to identify ASNA-1 mutations resistant to Retro-2.
DOI: https://doi.org/10.7554/eLife.48434.015
**Figure supplement 1.** Replication and validation of CRISPR-X screen.
DOI: https://doi.org/10.7554/eLife.48434.013
**Figure supplement 2.** Characterization of clonal ASNA1 A149V cells.
DOI: https://doi.org/10.7554/eLife.48434.014

populations (*Figure 4B*; *Figure 4—figure supplement 1A*; *Figure 4—source data 1*). The most common ASNA1 mutation in both replicates, C460T, results in an alanine to valine coding change (A149V), and we obtained two lines of cell-based evidence that it represents a bona fide resistance allele (*Figure 4C*).

First, we introduced the A149V mutation at the endogenous ASNA1 locus of a Cas9-expressing K562 line using CRISPR-mediated homologous recombination (*Figure 4—figure supplement 1B*). Using both mixed pools of edited cells (*Figure 4—figure supplement 1B*) and clonally selected homozygous mutants (*Figure 4—figure supplement 2A,B*), we found that the A149V mutation conferred resistance to the toxicity of DHQZ36.1, whereas a synonymous control mutation (C461T; A149A) did not (*Figure 4—figure supplements 1C* and *2C*). More critically, the A149V ASNA1 mutation also conferred resistance to the protective activity of Retro-2 against ricin whereas the synonymous mutation had little effect (*Figure 4D*; *Figure 4—figure supplements 1D* and *2D*).

Second, we analyzed the effects of Retro-2 and its analogs on the GFP-2A-RFP-SEC61B$_{TMD}$ reporter expressed in K562 cells. As expected from our previous reporter analysis in HEK293T cells, both Retro-2 and DHQZ36.1 but not the inactive DHQZ5 analog caused a robust decrease in the RFP:GFP ratio in both unedited wildtype and homozygous A149A cells (*Figure 4E*, *Figure 4—figure supplement 2E,F*). We noted no significant effect of the A149V mutation on ASNA1 protein level but detected in vehicle treated samples evidence of partial loss of function (*Figure 4E*; *Figure 4—figure supplement 2B,E*). However, analysis of the reporter dose response revealed that cells expressing ASNA1 A149V have an approximately ten times higher half maximal effective concentration (EC$_{50}$) for both Retro-2 and DHQZ36.1 than wildtype and ASNA1 A149A cells (*Figure 4E*, *Figure 4—figure supplement 2E*). We obtained similar results by ASNA1 complementation analysis in *ASNA1$^{KO}$* HEK293T cells following transient transfection (*Figure 4—figure supplement 2G*). In sum, our analysis of the A149V ASNA1 mutant demonstrates that the allele is resistant to the ability of DHQZ36.1 to inhibit cell growth, the ability of Retro-2 to protect cells from ricin, and the ability of Retro-2 to disrupt ER targeting of a TA reporter. These data provide further support for our model that Retro-2/DHQZ36.1 interferes with the TRC pathway, possibly by inhibiting ASNA1 directly.

## DHQZ36.1 directly inhibits TA protein delivery to ASNA1

As the final test of our MOA model, we investigated whether DHQZ36.1 disrupts TA protein delivery to ASNA1 using an established cell-free assay (*Guna et al., 2018*; *Mariappan et al., 2010*). Our model TA protein substrate comprised the TMD of STX5 appended to the SEC61B N-terminal domain (STX5$_{TMD}$), which facilitated subsequent chemical crosslinking analysis. Following in vitro translation in crude rabbit reticulocyte lysate (RRL), we monitored formation of the ASNA1~STX5$_{TMD}$ complex by ASNA1 immunoprecipitation after enrichment by size-separation on a sucrose gradient and stabilization by chemical crosslinking with bismaleimidohexane (BMH) (*Figure 5A*). DHQZ36.1 caused a small decrease in the formation of the crosslinked ASNA1×STX5$_{TMD}$ adduct (*Figure 5B,C*; *Figure 5—figure supplement 1A,B*). In contrast, DHQZ5, the inactive Retro-2 analog (*Carney et al., 2014*), behaved like the vehicle control. We also repeated our crosslinking analysis on SEC61B, a

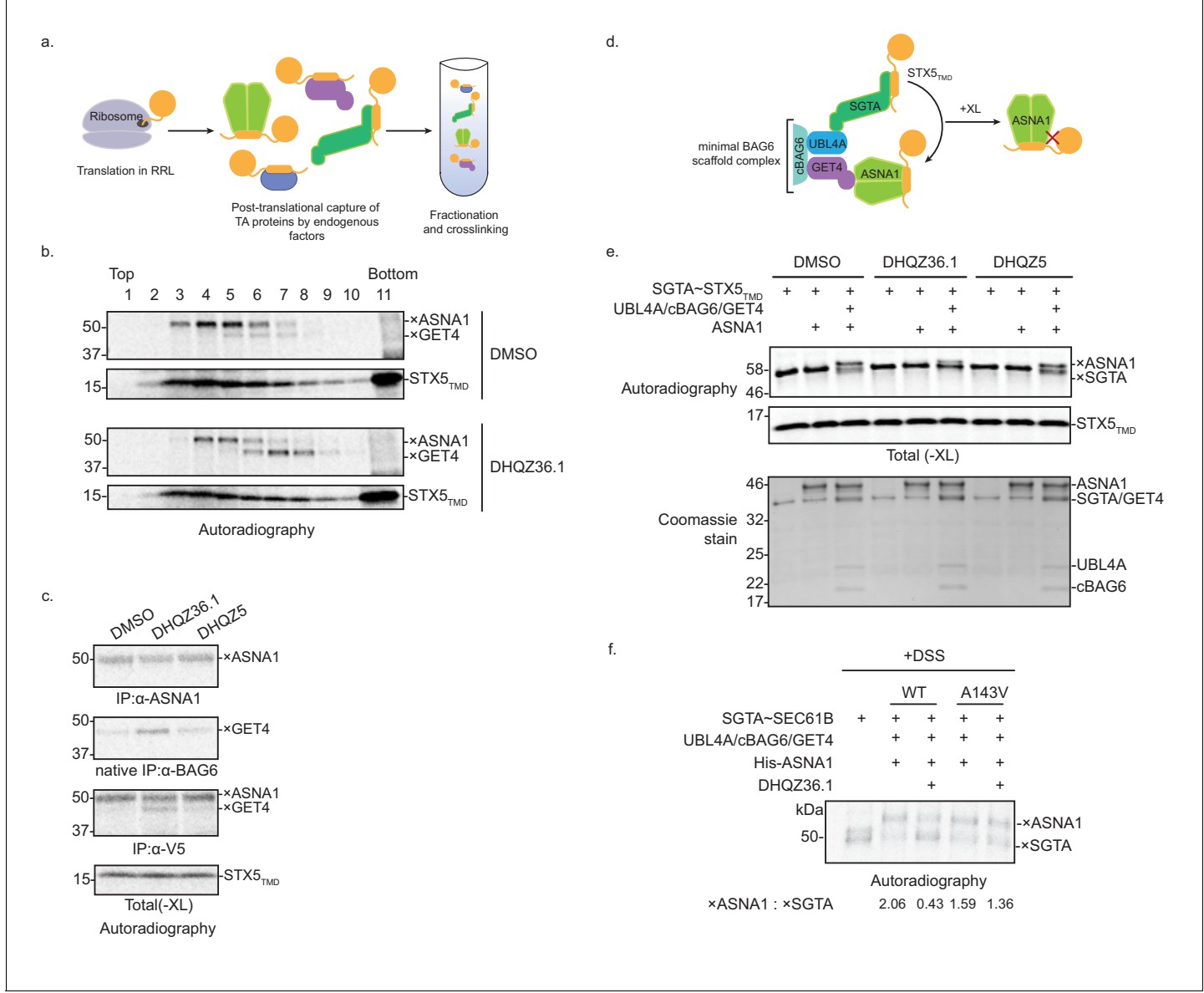

**Figure 5.** DHQZ36.1 blocks substrate transfer from SGTA to ASNA1 in vitro. (**A**) Schematic of the fractionation and crosslinking assay to monitor the cytosolic interactions of in vitro translated TA proteins. After translation in crude rabbit reticulate lysate (RRL), the TA proteins are captured by endogenous factors. (**B**) In vitro translation of a model TA substrate comprising of the cytosolic domain of SEC61B, the TMD of STX5 and a C-terminal V5 epitope (STX5$_{TMD}$) in the presence of $^{35}$S-methionine and 15 µM DHQZ36.1 in RRL. Completed lysate reactions were subjected to size fractionation by centrifugation in a 5–25% sucrose gradient and their individual fractions chemically crosslinked with 0.250 mM bismaleimidohexane (BMH). Samples were resolved by SDS-PAGE and visualized by autoradiography. Adducts to STX5$_{TMD}$ are denoted with ×. (**C**) $^{35}$S-methionine-labeled model TA substrate STX5$_{TMD}$, which has a C-terminal V5 epitope, was translated in crude RRL lysate in the presence of 15 µM of the indicated compounds. Translation reactions were subjected to chemical crosslinking (XL) with 0.250 mM BMH. Non-crosslinked samples were directly analyzed by SDS-PAGE. Crosslinked adducts to the TA protein substrate or ASNA1 were analyzed after denaturing immunoprecipitation (IP) with α-V5 or α-ASNA1 antibodies, respectively. For crosslinked products to GET4, samples were subjected to a non-denaturing IP with α-BAG6 antibody, which maintains the BAG6/GET4 interaction. Samples were visualized by autoradiography. Adducts to STX5$_{TMD}$ are denoted with ×. (**D**) Schematic of the TA protein transfer assay used in part e). Model TA substrate STX5$_{TMD}$ bound to SGTA is scaffolded into proximity with ASNA1 by a minimal BAG6 complex comprising the indicated components (see *Shao et al., 2017* for more details). Substrate transfer to ASNA1 is detected by chemical crosslinking (XL) with BMH. (**E**) Purified SGTA bound to STX5$_{TMD}$ (SGTA~STX5$_{TMD}$) and indicated recombinant TRC pathway components were incubated in the presence of 15 µM of DHQZ36.1 or DHQZ5 or mock treated (DMSO). Completed reactions were subjected to chemical crosslinking (XL) with 0.250 mM BMH. Crosslinked samples were resolved by SDS-PAGE and visualized by autoradiography (top) and Coomassie blue staining (bottom). Adducts to STX5$_{TMD}$ are denoted with ×. (**F**) The complex of $^{35}$S-methionine labeled SEC61B and SGTA (SGTA~SEC61B) was generated using the PURE system. Transfer reactions were assembled with wildtype ASNA1 or the putative Retro-2-resistant mutant of ASNA1 (A143V) in the presence of 1 µM DHQZ36.1. Substrate transfer to

*Figure 5 continued on next page*

*Figure 5 continued*

wildtype and mutant ASNA1 was detected by chemical crosslinking with disuccinimidyl suberate (DSS). Adducts to SEC61B are denoted with × and quantified using ImageJ.

DOI: https://doi.org/10.7554/eLife.48434.016

The following figure supplements are available for figure 5:

**Figure supplement 1.** In vitro analysis of DHQZ36.1's effect on ER targeting.

DOI: https://doi.org/10.7554/eLife.48434.017

**Figure supplement 2.** Biochemical analysis of mutant ASNA1.

DOI: https://doi.org/10.7554/eLife.48434.018

classic ASNA1 substrate in vitro, and found DHQZ36.1 treatment also reduced ASNA1~SEC61B complex formation. (*Figure 5—figure supplement 1C,D*).

Formation of ASNA1-TA protein complexes is preceded by TMD capture by SGTA, the most upstream factor in the TRC pathway. Following this pre-targeting step, TA proteins are transferred from SGTA to ASNA1 by a mechanism mediated by the BAG6 complex (comprising BAG6, GET4, and UBL4A) (*Shao et al., 2017*). We thus hypothesized that DHQZ36.1 interferes with TA protein delivery to ASNA1 from SGTA. This could explain our in vitro observation of the accumulation of another crosslinked adduct in the presence of DHQZ36.1 (*Figure 5B,C*, *Figure 5—figure supplement 1B,C,D*) that based on its molecular weight and native interaction with BAG6, could result from enhanced substrate crosslinking to GET4 (*Shao et al., 2017*; *Mariappan et al., 2010*). To test this hypothesis, we used an established in vitro assay with entirely purified components, including the close zebrafish homolog of ASNA1 (*Shao et al., 2017*). Following in vitro translation of STX5$_{TMD}$ in the presence of recombinant SGTA in the *E. coli* PURE system, we isolated SGTA-bound to STX5$_{TMD}$ by sucrose fractionation (*Figure 5—figure supplement 1E*). These complexes were then briefly incubated in the presence of ASNA1 and a minimal BAG6 scaffold complex (UBL4A/cBAG6/GET4) (*Figure 5D*) (*Shao et al., 2017*). Crosslinking analysis revealed robust formation of ASNA1×STX5$_{TMD}$ adducts that were dependent on addition of the scaffold complex (*Figure 5E*). Consistent with our hypothesis, the presence of DHQZ36.1 but not its biologically inactive derivative DHQZ5 (*Carney et al., 2014*) obstructed STX5$_{TMD}$, as well as SEC61B, delivery to ASNA1 under these conditions (*Figure 5E*; *Figure 5—figure supplement 1F*). Finally, we analyzed the effect of the zebrafish ASNA1 A143V mutation, which corresponds to the human ASNA1 A149V mutation that confers Retro-2 resistance (*Figure 5—figure supplement 2A*). While SEC61B was transferred efficiently to ASNA1 A143V, we no longer observed inhibition of ASNA1×TA protein adducts by addition of DHQZ36.1 (*Figure 5F*; *Figure 5—figure supplement 2B*). These data provide support for a more refined MOA model in which Retro-2 analogs directly interfere with the TRC pathway at the pre-targeting step to cause cytoprotection from ricin.

## Discussion

Here, we have demonstrated that Retro-2 and its hyperactive analog, DHQZ36.1, target the TRC pathway. Based on similarities between their genetic profiles (*Figure 1*), we hypothesized that Retro-2 acts via inhibition of TRC pathway members. Our subsequent findings provide three supporting lines of evidence for this model. First, DHQZ36.1 inhibits transfer of TA proteins from SGTA to ASNA1 in a biochemically defined system (*Figure 5*). Second, Retro-2 and DHQZ36.1 prevent the insertion of newly-synthesized TA protein reporters (*Figures 2–3*). Third, an ASNA1 mutation prevents four distinct forms of Retro-2/DHQZ36.1 activity: cytoprotection against ricin, inhibition of growth, ASNA1 substrate destabilization associated with ER targeting disruption, and inhibition of assisted substrate transfer from SGTA to ASNA1.

Our Retro-2 mechanism of action model also provides a satisfying explanation for two previous observations. First, Retro-2 treatment interferes with STX5 localization to the Golgi (*Stechmann et al., 2010*), which can now be understood as an expected consequence of ASNA1 inhibition. Second, both Retro-2 treatment and ASNA1 knockout reduce insulin content and induce ER stress in isolated mouse islet cells (*Norlin et al., 2016*). Our work lends a unifying mechanistic explanation for this phenotypic commonality. Beyond this, determining in greater detail how Retro-2 blocks the pre-targeting step of the TRC pathway is an important future goal. Intriguingly, mapping

of our top ASNA1 resistance allele onto the structure of yeast Get3 revealed its location within Helix 6 (*Figure 5—figure supplement 2A*). These structural elements appear to have an allosteric role in stimulating ATPase activity (*Mateja et al., 2009*) and warrant an enzymatic analysis of the effects of Retro-2 on ASNA1 during the pre-targeting step.

The ability of Retro-2 to inhibit the TRC pathway and disrupt STX5 may be sufficient to explain its protective activity against ricin and other toxins and viruses that similarly rely on retrograde trafficking (*Nelson et al., 2013*; *Lipovsky et al., 2013*; *Maru et al., 2017*; *Nonnenmacher et al., 2015*). However, retrograde trafficking has not been implicated in the life cycles of many Retro-2-sensitive viruses and endocytic pathogens such as human cytomegalovirus (*Cruz et al., 2017*), filoviruses such as Marburg and Ebola (*Shtanko et al., 2018*), Vaccinia (*Sivan et al., 2016*), chlamydia (*Herweg et al., 2016*), leishmania (*Craig et al., 2017*), and enteroviruses (*Dai et al., 2017*). In those instances, we speculate that either functions of STX5 beyond retrograde trafficking or other clients of the TRC pathway are responsible for the protective activity of Retro-2. In addition, our work provides impetus to examine the efficacy of Retro-2 against viruses such as Herpes simplex virus 1 (*Ott et al., 2016*), Epstein-Barr (*Horst et al., 2011*), and human papillomavirus 6 and 8 (*Sullivan and Coscoy, 2010*), which are known to depend on the TRC pathway.

In our paired-gene CRISPRi strategy, the top four genes were all members of the TRC pathway. The fifth gene was SRP54, a factor that plays an analogous role to ASNA1 in the co-translational ER protein targeting pathway (*Figure 1C*). We have found no evidence, however, that Retro-2 affects co-translational ER protein targeting (*Figure 2D*). While very low levels of interaction between TA proteins and SRP54 have been detected in vitro (*Stefanovic and Hegde, 2007*), a wealth of evidence demonstrates that the SRP pathway does not mediate TA protein insertion under these conditions (*Stefanovic and Hegde, 2007*; *Kutay et al., 1995*). Nonetheless, SRP54 is capable of efficiently recognizing a TA protein TMD sequence when the latter is artificially presented to it during protein translation by the ribosome. Thus, the genetic profile of SRP54 possibly reflects disruption of the TRC pathway by post-translational engagement with TMD regions of orphaned SRP clients in SRP54 knockdown cells.

Together, our work demonstrates the use of paired-gene CRISPR screening technology in mammalian cells to translate the chemical-genetic signature of Retro-2 into a testable model for its MOA that explains its ability to protect cells against ricin. Additionally, it illustrates a powerful and generalizable approach for model validation by targeted genomic mutagenesis of an MOA candidate using dCas9-targeted AID (CRISPR-X). Together, this combination of technologies charters a new path for direct mechanistic dissection of novel compounds in human cells. Finally, this study defines the first known chemical inhibitor of the TRC pathway and could be an impetus for pharmaceutical efforts to drug this pathway as a therapy against diverse viruses and pathogens.

## Materials and methods

### Cell culture and cell lines

Cell culture was carried out as previously described (*Deans et al., 2016*). K562 cells (ATCC) were cultured in RPMI 1640 (Gibco) media and supplemented with 10% FBS (Hyclone), pen-strep (10,000 I. U./mL), and L-glutamine (2 mM). HeLa cells (ATCC) were cultured in DMEM (Gibco) media and supplemented with 10% FBS (Hyclone), pen-strep (10,000 I.U./mL), and L-glutamine (2 mM). These cell lines were maintained in a controlled humidity incubator at 37° C with 5% $CO_2$. HEK293T cells were grown in a standard water-jacketed incubator with 5% $CO_2$ and maintained in DMEM (Gibco) media supplemented with 10% FBS (ATCC) and penicillin-streptomycin (10,000 U/mL) (Thermo Fisher Scientific). All cell lines were passaged less than 25 times passaged by trypsinization with 0.05% Trypsin-EDTA (Life Technologies).

### Chemicals and drugs

Retro-2, DHQZ5, and DHQZ36.1 were chemically synthesized as previously described (*Nelson et al., 2013*; *Craig et al., 2017*; *Carney et al., 2014*). Stock solutions were prepared in dimethylsufloxide (DMSO).

## Antibodies

α-ASNA1, α-SGTA and α-BAG6 antibodies were gifts from Susan Shao. α-GET4 and α-SQSTM1 antibodies were obtained from Abcam. α-alpha tubulin antibody (YOL1/34) and α-V5 antibody were obtained from Santa Cruz antibodies and Invitrogen, respectively. α-STX5 was obtained from Synaptic Systems (110053). Secondary antibodies were goat α-mouse-IgG-HRP (Biorad), goat α-rabbit-IgG-HRP (Biorad), goat α-mouse-IgG-Cy5 (Thermo Fisher Scientific), goat α-rabbit-IgG-Cy3 (Thermo Fisher Scientific), goat α-rat 488 Alexa Fluor (Thermo Fisher Scientific), goat α-rabbit 488 Alexa Fluor (Thermo Fisher Scientific), and goat α-mouse 647 Alex Fluor (Thermo Fisher Scientific).

## Vectors

LentiCRISPR v2 was a gift from Feng Zhang (Addgene plasmid #52961). pCW57-GFP-2A-MCS was a gift from Adam Karpf (Addgene plasmid #71783). pLV-azurite was a gift from Pantelis Tsoulfas (Addgene plasmid #36086). pET29b was obtained from Novagen. pCS470 tfLC3, pCS625 BFP, psPAX, and pVSV were gifts from Christopher Shoemaker. pGEX-6P1 GST-SGTA was a gift from Robert Keenan. pGEX GST-GET4, pACYC His-cBAG6[1004–1132], pRSETA His-UBL4A, pET28a zebrafish His-ASNA1 and PUREExpress 3HA-SEC61B were gifts from Susan Shao. The vector for SEC61B-3F4 was a gift from Ramanujan Hegde.

## Single-gene CRISPRi library construction, screening, and analysis

A sgRNA library targeting 288 genes involved in ricin resistance were chosen from a previously designed CRISPRi library (*Horlbeck et al., 2016*) along with 2000 non-targeting and safe-targeting controls from a separate design (*Morgens et al., 2017*). As previously described (*Morgens et al., 2017*), sgRNAs were synthesized (Agilent) and transformed into mCherry positive, puromycin resistant, third generation, lentiviral plasmids driven by an mU6 promoter (Addgene pMCB320). Lentivirus was created using these plasmid libraries by PEI transfection into HEK293T cells along with packaging vectors. Virus was then spin infected into dCas9-KRAB expressing K562 cells (*Gilbert et al., 2014*) at 500× guide coverage and low MOI. Three days post-infection, cells were selected with puromycin (1 µg/mL; Sigma) for an additional three days. Cells were then split into four conditions with two replicates each at 1000× guide coverage. Cells were then pretreated with 10 µM Retro-2 or DMSO for 24 hr and remained in Retro-2 for the entire course of the experiment. Ricin treatment (1.25 ng/mL; Vector Labs) was carried out for 24 hr initially, followed by a three day recovery before being treated again (2.5 ng/mL) for 24 hr. Cells were again allowed to recover for three days before receiving a third dose of ricin at 2.5 ng/µL for 24 hr. Untreated and Retro-2 treated replicates were split each day to maintain cells at 500,000 cells per mL. Four days after the last third ricin pulse and 13 days after the pretreatment, all cells were spun into fresh media and expanded before being spun down for DNA extraction. Genomic DNA was extracted using QIAamp DNA mini kits, guides were amplified as previously described (*Deans et al., 2016*), and libraries were sequenced on a NextSeq 550. Gene-level effects and scores were then generated using casTLE version 1.0 (*Morgens et al., 2016*), a maximum likelihood estimator that aggregated information from multiple sgRNAs targeting the same gene into an interpretable gene-level effect size and associated significance. Scripts are available at https://bitbucket.org/dmorgens/castle (copy archived at https://github.com/elifesciences-publications/Source-code-for-casTLE-analysis).

## Paired-gene CRISPRi library construction, screening, and analysis

Three sgRNAs each for 105 genes and 50 safe-targeting sgRNAs (*Morgens et al., 2017*) were selected, and a paired-guide library was cloned as previously described (*Han et al., 2017*). Briefly, two libraries of these were synthesized and cloned into distinct mU6 and hU6 containing plasmids. The two plasmid pools were then combined in an all-by-all cloning step, creating a pool of mU6 and hU6 containing plasmid each containing two guides. This library was then lentivirally installed into a K562 line stably expressing dCas9-KRAB fusions as above (*Gilbert et al., 2014*). Cells were allowed to recover for three days before selection with puromycin (1 µg/mL; Sigma) for three additional days. Cells were then split into four pools, and treated in duplicate with ricin or left untreated as above. After two weeks and three treatments, cells were frozen down and DNA was extracted using QIAGEN maxi-prep kits. The double-sgRNA vector was amplified from the genome and sequenced

on a NextSeq 550 as previously described (*Han et al., 2017*). Double-gene level effects were obtained using casTLE version 1.0 (*Morgens et al., 2016*) by first combining counts for reciprocal pairs (i.e. guideA_guideB and guideB_guideA) and the nine such pairs considered as independent measurements with all safe_safe pairs considered as negative controls.

## Generation of the vectors for two-color reporters and in vitro translation

RFP was PCR amplified from pCS470 tfLC3 (*Shoemaker et al., 2017*) and subcloned into the MluI/BamHI sites of pCW57-GFP-2A-MCS to yield pCW57-GFP-2A-RFP The coding region of SEC61B protein was PCR amplified from the vector for SEC61B-3F4, and subcloned into the BamHI/NdeI sites of a pET29b backbone with C-terminal opsin, V5 and His tags. These vectors served as templates for amplifying the TMD with a C-terminal opsin tag and subcloning into the SgrAI/BamHI sites of pCW57-GFP-2A-RFP. To generate pCW57 GFP-2A-RFP-SEC61BTMD-BFP, BFP was PCR amplified from pLV-azurite and appended via a short GGGS linker to the opsin tag by overlap extension PCR.

The model TA protein substrate with the TMD of STX5 consists of the cytosolic domain of SEC61B, the TMD of STX5 and a C-terminal V5 epitope. To make the vector for SEC61B-STX5$_{TMD}$ for in vitro translation in rabbit reticulocyte lysate, the coding region of the cytosolic domain of SEC61B was appended to the TMD of STX5 by overlap extension PCR and then subcloned into the BamHI/NdeI sites of a pET29b backbone with C-terminal opsin, V5 and His tags to yield pET29b SEC61B-STX5$_{TMD}$. To make the vector for in vitro translation of SEC61B-STX5$_{TMD}$ and SEC61B in the PURE system, SEC61B-STX5$_{TMD}$ and SEC61B was PCR amplified from pET29b SEC61B-STX5$_{TMD}$ and pET29b SEC61B, respectively, and subcloned into the NdeI/BamHI sites of the DHFR PUREExpress Control Template (New England Biolabs).

## Generation of ASNA1$^{KO}$ HEK293T cells

Oligos encoding a guide RNA against ASNA1 (CTGAAGTGGATCTTCGTCGG) were cloned into the BsmBI site of lentiCRISPRv2. HEK293T cells were transfected using lipofectamine 3000. Cells were allowed to recover for 48 hr and then treated with puromycin (2 µg/mL) for 24 hr to select for transfected cells. Puromycin was removed and cells were passaged for another 72 hr. Cells were diluted by limiting dilution into 96-well plates to generated clonal isolates. Colonies were allowed to grow up for ~10 days. Wells were scored for the presence of single colonies and single-colony wells were propagated. Knockout cells were confirmed by immunoblot against ASNA1. Indel formation in the *ASNA1$^{KO}$* cells was confirmed by T7 endonuclease assay and standard commercial Sanger sequencing (Eton Bioscience). In brief, genomic DNA (gDNA) was extracted using QuickExtract DNA extraction solution (Lucigen) according to manufacturer's instructions. The ASNA1 locus was amplified using the primers ASNA1 KO confirm F4 (CTGGAGCCTACACTTAGCAAC) and ASNA1 KO confirm R1 (CGCCCAGTGGTATATCCTAC). The resulting amplicon was purified using a Zymoclean Gel DNA Recovery Kit (Zymo Research) and either hybridized and treated with T7 endonuclease I (New England Biolabs) according to manufacturer's instruction or cloned into pJET1.2/blunt with the CloneJET PCR Cloning Kit (Thermo Fisher Scientific), and sequenced at Eton Bioscience.

## Lentiviral generation

Wildtype cells were grown to 90% confluency in Gibco Opti-MEM reduced serum media (Thermo Fisher Scientific) supplemented with 5% FBS. Cells were transfected with 1 µg psPAX, 0.25 µg pVSV, 0.75 µg vector using lipofectamine 3000 (Thermo Fisher Scientific) and incubated for >6 hr, at which time the media was replaced with fresh Opti-MEM reduced serum media. The supernatant was harvested twice over two days and pooled together. Cell debris was removed by centrifugation and the lentivirus was stored at −80 ˚C.

## Viral transduction of HEK293T cells

Cells were grown to 75–90% confluent in a 12-well plate and transduced with lentivirus overnight in DMEM supplemented with 10% FBS and 8 µg/mL polybrene (Sigma) but lacking penicillin/streptomycin. Media was exchanged to remove polybrene. Cells were allowed to grow for an additional 24 hr prior to selection with puromycin (2 µg/mL).

## Flow cytometry analysis

Cells expressing two-color reporters were grown in a 6-well culture plate and pre-treated with indicated compound or mock-treated with DMSO for 1 hr followed by doxycycline hyclate (final concentration of 250 ng/mL) (Sigma) for approximately 18 hr. Cells were detached, pelleted, and resuspended with ice-cold phosphate buffered saline (10 mM $Na_2HPO_4$, pH 7.4, 1.8 mM $KH_2PO_4$, 2.7 mM KCl, 137 mM NaCl) (PBS), and analyzed by flow cytometry using a BD LSRII (BD Biosciences). Data were analyzed by FlowJo (FlowJo, LLC) and R. In particular, the dose response curves were analyzed with the drc package in R.

## Automated Microscopy

shRNAs targeting ASNA1 or scrambled shRNA controls derived from a previous library design (*Kampmann et al., 2015*) were stably expressed in wildtype HeLa cells using lentivirus. Cells were plated in 24-well plates at 25,000 cells per well and pretreated for 1 hr at the appropriate drug concentration. They were then treated for 24 hr with 2.5 ng/mL ricin (Vector Labs). Ricin media was replaced with fresh media with the appropriate drug concentration along with 50 nM CellTox Green. Cells were then imaged on an Incucyte Zoom for 72 hr, imaging every 4 hr. Confluence was determined from phase images using Incucyte software.

## Confocal microscopy

HeLa cells stably expressing shASNA1 or scrambled control shRNAs were plated at 50,000 cells per well in glass bottom plates and treated either 10 µM Retro-2 or 3 µM DHQZ36.1 for 24 hr. Cells were fixed in 4% PFA for 10 min, permeabilized in 0.3% Triton-X for 10 min, and blocked in fresh 3% BSA for >1 hr. Fixed cells were stained with primary antibodies for 1 hr (1:300 STX5-rabbit; 1:250 GM130-mouse) and secondary antibodies for 1 hr (1:2000 anti-mouse-488; 1:2000 anti-rabbit-647). After high-volume Dulbecco's phosphate-buffered saline (DPBS) (Thermo Fisher Scientific) washes, cells were mounted with VectaShield w/DAPI. Cells were imaged using an inverted Nikon Eclipse Ti confocal microscope with an oil immersion objective (Plan Apo, numerical aperture (NA)−1.5, 60× or 100×, Nikon), and an Andor Ixon3 EMCCD camera or Andor Zyla sCMOS camera. All images presented are max-projections using FIJI (*Schindelin et al., 2012*) from 0.2 µm Z-stacks. Image analysis was carried out using csth imaging package and pyto segmenter (*Weir et al., 2017*).

## CRISPR-X mutagenesis

172 guides tiling the ASNA1 coding sequence were obtained from CHOP-CHOP (*Montague et al., 2014*), synthesized (Agilent), and cloned into an mU6 driven guide expression vector. Lentivirus was produced as above and the tiling library was infected into an N-terminal fusion, CRISPR-X (dCas9-XTEN-AID*Δ; unpublished; GTH) expressing K562 at low MOI. This construct is an N-terminal fusion to a nuclease dead version of Cas9 (dCas9) of a hyperactive version of Activation-Induced cytidine Deaminase (AID*) (*Wang et al., 2009*) with the nuclear exclusion sequence removed (AID*Δ) with the XTEN linker (*Schellenberger et al., 2009*). Three days post-infection, cells were selected using 1 µg/mL puromycin. Cells were then treated for ~4 weeks with 20 µM DHQZ36.1. RNA was extracted from cells using Qiagen RNeasy mini kits, and an ASNA1-specific primer were used for first-strand synthesis (Roche AMV) and PCR. The isolated ASNA1-cDNA was then sequenced using Nextera XT on a NextSeq 550. Reads were aligned using BWA (*Li and Durbin, 2009*), and variants were called using a custom script (*Hess et al., 2016*). For each base with a minimum of 500× read coverage, a percent allele frequency was calculated. Allele frequency from pre-selected cells was measured as above from cells after puromycin selection but before DHQZ36.1 treatment.

## ASNA1 allele installation and validation

1 µg of 100 basepair ssDNA ultramer template (IDT) containing the C460T mutation and 2 µg a mU6 driven sgRNA targeting ASNA1 were electroporated into Cas9-expressing K562 using the T-175 program with Lonza Nucleofecter 2b. A safe-targeting guide, a template for C461T, a no-template control, and a mock transfection were included as controls. After three days, cells were sorted for guide expression by GFP. After four more days, cells were sorted for dark to eliminate random integrants. To monitor resistance to DHQZ36.1 toxicity, cells were treated with 20 µM DHQZ36.1 or DMSO for 4 days and viability was monitored via cytometry. To monitor resistance to Retro-2's

protective activity, cell were pretreated with 10 μM Retro-2 or DMSO for 24 hr, then treated with 3.75 ng/mL ricin (Vector Labs) for 24 hr. The toxin was spun out and viability and cell number were measured after three days via cytometry.

To generate clonal mutant lines, cells were electroporated as above. Cells receiving a safe-targeting guide, template for C460T, and a template for C461T, were sorted for GFP expression after three days. After four more days, dark cells were single-cell sorted into 96-well plates and allowed to recover over the next 2–3 weeks. DNA was isolated from the clones that grew, and editing was monitored by PCR and Sanger sequencing. Homozygous mutants were then selected and treated as above.

## Viral transduction and flow cytometry analysis of K562 cells

Clonal wildtype, ASNA1 A149V (C460T), and ASNA1 A149A (C461T) K562 cells were incubated with lentivirus in RPMI 1640 supplemented with 10% FBS, 1% L-Glutamine and 8 μg/mL polybrene (Sigma) but lacking penicillin/streptomycin. Half a million cells of each cell line were spin-infected at 800 g at 32 °C for 30 min. Cells were allowed to grow for an additional 24 hr prior to selection with puromycin (1 μg/mL). Transduced cells were then treated with doxycycline hyclate (final concentration of 1 μg/mL) for approximately 20 hr. Cells were washed and resuspended with ice-cold PBS and bulk sorted using a MoFlo XDP cell sorter (Beckman Coulter) for GFP and RFP positive cells.

These cells were grown in 6-well plate and then pre-treated with the indicated compounds for 1 hr followed by induction with doxycycline hyclate (500 ng/mL) for ~18 hr. Cells were harvested and resuspended in ice-cold PBS and analyzed by flow cytometry using BD LSRII (BD Biosciences). Data were analyzed by FlowJo (FlowJo, LLC) and R.

## Generation of the expression vectors of BFP-ASNA1, BFP-ASNA1 A149V, and BFP-ASNA1 D74N

ASNA1 was PCR amplified from human ASNA1 (Accession: BC002651) cDNA from the Mammalian Gene Collection (Dharmacon) and subcloned into the NdeI/XhoI sites of pET29b. Point mutations were introduced using QuikChange site-directed mutagenesis with *PfuTurbo* DNA polymerase (Aligent). To N-terminally tag ASNA1 with BFP, BFP and ASNA1 alleles were PCR amplified from pCS625 BFP and pET29b ASNA1 vectors, respectively, and inserted into the EcoRV/BamHI site of pCS625 BFP by Gibson assembly. Primer sequences were used to introduce a GGGS linker separating ASNA1 from BFP.

## Two-color reporter analysis of A149V BFP-ASNA1

Wildtype and $ASNA1^{KO}$ GFP-2A-RFP-SEC61B$_{TMD}$ HEK293T cells were grown in a 6-cm plate and treated with doxycycline hyclate (final concentration of 250 ng/mL) (Sigma) for approximately 20 hr. Cells were detached, pelleted, and resuspended with ice-cold PBS and bulk sorted using a MoFlo XDP cell sorter (Beckman Coulter) for GFP and RFP positive cells. The latter cells were then grown in a 6-cm plate and transfected with 2 μg of vectors expressing BFP-ASNA1 variants or BFP alone using lipofectamine 3000. After approximately 40 hr, cells were allowed to recover in fresh Gibco Opti-MEM reduced serum media (Thermo Fisher Scientific) for 1 hr before incubation with Retro-2 (10 μM) or mock-treatment with DMSO for an additional hour followed by treatment with doxycycline hyclate (250 ng/mL; Sigma) for 24 hr. Sorted but non-transfected wildtype control cells were grown and treated in parallel. Cells were detached, pelleted, and resuspended with ice-cold PBS, and analyzed by flow cytometry using a BD LSRII (BD Biosciences). Data were analyzed by FlowJo (FlowJo, LLC) and R.

## Recombinant protein purification

GST-tagged SGTA, zebrafish His-tagged ASNA1, GST-tagged GET4, His-tagged cBAG6, and His-tagged UBL4A were purified from BL21-CodonPlus (DE3)-RIPL or BL21 (DE3) competent *E. coli* cells as previously described (*Shao et al., 2017*). In brief, cells were transformed with the expression plasmids encoding the protein and were grown in LB media at 37 °C under the appropriate antibiotic selection. For GST-tagged SGTA, zebrafish His-tagged ASNA1 and His-tagged UBL4A, cells were grown to $A_{600}$ = 0.4–0.6 and expression was induced with 0.1–0.2 mM of isopropyl β-D-1-thiogalactopyranoside (IPTG) for 2–3 hr. For GST-tagged GET4 and His-tagged cBAG6, cells were grown to

$A_{600}$ = 0.7–0.9, cooled and expression was induced with 0.2 mM of IPTG overnight at 16 °C. Cells were then harvested.

For ASNA1, cells were resuspended in Tris lysis buffer (50 mM Tris, pH 7.5, 300 mM NaCl, 2% glycerol, 2 mM beta mercaptoethanol (BME)) supplemented with cOmplete Protease Inhibitor Cocktail and phenylmethane sulfonyl fluoride (PMSF). Cells were lysed by high pressure homogenization by two passes through an EmulsiFlex-C3 (Avestin, Inc). The cell lysate was clarified by centrifugation and bound to a Ni-NTA resin (Thermo Fisher Scientific) column by gravity flow. Columns were washed and then eluted with 50 mM Tris, pH7.5, 150 mM NaCl, 2% glycerol, 2 mM BME, 250 mM imidazole. Protein was desalted with Econo-Pac 10DG Desalting Prepacked Gravity Flow Columns (Bio-Rad) into 50 mM Tris 7.5, 50 mM NaCl, 2 mM BME and 2% glycerol and stored −80 °C until use.

For SGTA, cells were resuspended in phosphate salt lysis buffer (10 mM $Na_2HPO_4$, pH 7.4, 1.8 mM $KH_2PO_4$, 2.7 mM KCl, 250 mM NaCl, 10 mM imidazole, 1 mM dithiothreitol (DTT)) and lysed by high pressure homogenization. The cell lysate was clarified by centrifugation and bound to a glutathione-sepharose (Millipore Sigma) column by gravity flow. Columns were washed and then eluted with 50 mM Tris, pH 8, 25 mM reduced glutathione. Peak elutions were determined by SDS-PAGE and Coomassie staining then pooled and dialyzed against 25 mM Hepes, pH 7.4, 150 mM KAcO, 10 mM imidazole, 10% glycerol, 1 mM DTT in the presence of PreScission Protease (Millipore Sigma). The protease and tags were removed from purified SGTA by passing the dialyzed protein through a glutathione-sepharose column. The flow through was collected and stored at −80 °C until use.

For the minimal scaffold complex (UBL4A/cBAG6/GET4), cells were resuspended in phosphate salt lysis buffer and lysed with two passages through the EmulsiFlex-C3. The cell lysates then were clarified by centrifugation. The cell lysates for GST-GET4 and His-tagged cBAG6 incubated together at 4 °C for 1 hr with shaking and were co-purified together by metal-affinity chromatography through a Ni-NTA resin column. His-tagged UBL4A was also bound to a Ni-NTA resin column by gravity flow. Columns were washed with phosphate salt lysis buffer and eluted with elution buffer (10 mM $Na_2HPO_4$, pH 7.4, 1.8 mM $KH_2PO_4$, 2.7 mM KCl, 250 mM NaCl, 250 mM imidazole, 1 mM DTT). Peak elutions were determined by SDS-PAGE and Coomassie staining, and combined for complex formation. Proteins were dialyzed against PBS supplemented with 5% glycerol and 1 mM DTT in the presence of acTEV protease (Thermo Fisher Scientific). The dialyzed material was clarified by centrifugation and the supernatant was bound to a glutathione-sepharose column by gravity flow. The complex was eluted with 50 mM Tris, pH 8, 25 mM reduced glutathione. Peak elutions were pooled together and dialyzed in 25 mM Hepes, pH 7.4, 150 mM KOAc, 10 mM imidazole, 10% glycerol and 1 mM DTT in the presence of PreScission Protease. The minimal scaffold complex was cleared of proteases and cleaved tags by subtraction through Ni-NTA and glutathione-sepharose columns. The minimal scaffold complex was stored at −80 °C until use.

## In vitro transcription

Templates for transcription were generated by PCR using a 5' primer containing the T7 promoter that anneals to the first ~30 bp of the open reading frame and a 3' primer containing the stop codon and polyA tail that anneals to the last ~30 bp of the open reading frame. PCR templates for transcription were used to make mRNAs by incubation with components of mMESSAGE mMACHINE T7 transcription kit (Thermo Fisher Scientific) at 37 °C for 2 hr. The mRNA was purified by acid phenol-chloroform extraction before being in vitro translated.

## In vitro translation, fractionation and crosslinking assay

SEC61B-STX5$_{TMD}$-V5 was synthesized from purified mRNA (5 ng of mRNA/10 μL of translation reaction) with prepared rabbit reticulocyte lysate (RRL) (*Stefanovic and Hegde, 2007*) supplemented with $Mg^{2+}$ and spermidine (final concentration of 1.2 mM and 0.4 mM, respectively) in the presence of DMSO or 15 μM DHQZ36.1. Translation was initiated with [35]S-methionine and mRNA and incubated for 30 min at 32 °C. For SEC61B-V5, translation was initiated with purified mRNA at 100 ng of mRNA/10 μL of translation reaction.

20 μL of the translation reaction was subjected to size fractionation through a 200 μL 5–25% sucrose gradient in physiological salt buffer (50 mM Hepes pH 7.4, 100 mM KOAc, 2 mM Mg(OAc)$_2$) (PSB). Samples were centrifuged for 145 min at 4 °C using a TLS-55 rotor (50 k rpm) with the slowest

acceleration and deceleration settings. Eleven 20 µL fractions were collected. Fractions were incubated with 0.250 mM bismaleimidohexane (BMH) for 1 hr on ice. Reactions were quenched with protein sample buffer and analyzed by SDS-PAGE and autoradiography.

To analyze the crude lysate by fractionation, 20 µL of RRL was directly subjected to size fractionation through a 200 µL 5–25% sucrose gradient in PSB. Samples were centrifuged for 145 min at 4 °C using a TLS-55 rotor (50 k rpm) with the slowest acceleration and deceleration settings. Eleven 20 µL fractions were collected and analyzed by SDS-PAGE and immunoblotted with α-BAG6, α-ASNA1, α-SGTA, and α-GET4.

## In vitro translation, crosslinking and immunoprecipitation

Following in vitro translation, reactions were diluted 10-fold in PSB and incubated on ice for 1 hr with 0.250 mM BMH. Crosslinked samples were split and treated with DTT and phenylmethane sulfonyl fluoride (PMSF). For denaturing immunoprecipitation (IP) of the substrate or ASNA1, samples were solubilized with the addition SDS-containing Tris buffer (final concentration of 1% SDS) and boiled. The samples were then diluted with ice-cold IP buffer (50 mM HEPES, pH 7.4, 150 mM NaCl, 1% Triton X-100) and incubated with α-V5 antibody or α-ASNA1 antibody and Protein G dynabeads or Protein A agarose, respectively, at 4 °C overnight. For native IP of Bag6-associated crosslinked products, samples were diluted with ice-cold IP buffer and incubated with α-BAG6 antibodies and Protein A agarose at 4 °C overnight. The beads were washed three times with IP buffer and then eluted with protein sample buffer. Samples were analyzed by SDS-PAGE and autoradiography.

## TA substrate transfer assay

SGTA~SEC61B-STX5$_{TMD}$ (SGTA~STX5$_{TMD}$) and SGTA~3 HA-SEC61B (SGTA~SEC61B) complexes were generated as previously described (*Shao et al., 2017*; *Mateja et al., 2015*). Briefly, the PURExpress vectors were directly transcribed and translated with the commercial PURExpress in vitro protein synthesis kit (New England Biolabs) supplemented with purified SGTA (final concentration of ~14 µM) for 90 min at 37 °C. 50 µL of the translation reaction was diluted to 200 µL with PSB and applied to a 2 mL 5–25% sucrose gradient in PSB. Gradient centrifugation was carried out for 5 hr at 4 °C using a TLS-55 rotor (55 k rpm) with the slowest acceleration and deceleration settings. Eleven 200 µL fractions were collected and fractions 3–5 containing SGTA~TA protein complexes were pooled and stored at −80 °C until used.

Substrate transfer reactions were assembled on ice by mixing SGTA~TA protein complexes with recombinant zebrafish ASNA1 and UBL4A/cBAG6/GET4 scaffold complex (all proteins were present at ~1 µM final concentration) in the presence of 1 mM ATP, 2 mM Mg(OAc)$_2$, and 15 µM DHQZ36.1 or DHQZ5. Transfer was initiated by warming up the reactions to 32 °C for 1 min before placing them back on ice and incubating them with 0.250 mM BMH for 1 hr. Crosslinking reactions were quenched in protein sample buffer and analyzed by SDS-PAGE on a 10% gel and autoradiography.

The A143V point mutation was introduced to His-tagged ASNA1 using QuikChange site-directed mutagenesis with *PfuTurbo* DNA polymerase (Aligent). Recombinant ASNA1 A143V was induced and purified by Ni-affinity chromatography. Substrate transfer reactions were performed as described above with one exception; to crosslink the proteins, transfer reactions were incubated with 0.2 mM disuccinimidyl suberate (DSS) (Thermo Scientific) for 30 min at room temperature. Crosslinking reactions were quenched in protein sample buffer and analyzed by SDS-PAGE on a 10% gel and autoradiography. ImageJ was used for quantification.

## Acknowledgements

We thank S Shao for reagents and scientific advice, F Wang and C Shoemaker for mentorship, S Dixon and members of the Denic, Sello, and Bassik labs for technical expertise and helpful discussions. This work was funded by the NIH Director's New Innovator Award Program (project no. 1DP2HD084069-01 to MCB), NIH/NHGRI (training grant T32 HG000044 to DWM). This material is based on work supported by the National Science Foundation Graduate Research Fellowship (grant DGE-114747 to DWM and ELH and DGE-1656518 to AL). This work was supported by the National Institute of Health under grant No. NIH/NIGMS 1R35GM127136 (VD) and by Harvard University and Brown University.

## Additional information

### Funding

| Funder | Grant reference number | Author |
|---|---|---|
| National Institutes of Health | DP2HD084069 | Michael C Bassik |
| National Institutes of Health | T32 HG000044 | David W Morgens |
| National Institute of General Medical Sciences | 1R35GM127136 | Vladimir Denic |
| National Science Foundation | DGE-114747 | David W Morgens Emma L Handy |
| National Science Foundation | DGE-1656518 | Adam Lavertu |

The funders had no role in study design, data collection and interpretation, or the decision to submit the work for publication.

### Author contributions

David W Morgens, Conceptualization, Data curation, Software, Formal analysis, Validation, Investigation, Visualization, Writing—original draft, Writing—review and editing; Charlene Chan, Conceptualization, Data curation, Formal analysis, Validation, Investigation, Visualization, Writing—original draft, Writing—review and editing; Andrew J Kane, Adam Lavertu, Formal analysis, Investigation; Nicholas R Weir, Data curation, Investigation; Amy Li, Michael M Dubreuil, C Kimberly Tsui, Nicole Polyakov, Emma L Handy, Philip Alabi, Amanda Dombroski, Investigation; Gaelen T Hess, Formal analysis, Investigation, Methodology; Kyuho Han, David Yao, Methodology; Jing Zhou, Investigation, Methodology; Russ B Altman, Formal analysis, Supervision; Jason K Sello, Conceptualization, Supervision, Funding acquisition, Methodology, Project administration, Writing—review and editing; Vladimir Denic, Conceptualization, Supervision, Funding acquisition, Writing—original draft, Project administration, Writing—review and editing; Michael C Bassik, Conceptualization, Supervision, Project administration, Writing—review and editing

### Author ORCIDs

Nicholas R Weir (iD) http://orcid.org/0000-0002-1797-849X
Vladimir Denic (iD) https://orcid.org/0000-0002-1982-7281
Michael C Bassik (iD) https://orcid.org/0000-0001-5185-8427

### Decision letter and Author response

Decision letter https://doi.org/10.7554/eLife.48434.021
Author response https://doi.org/10.7554/eLife.48434.022

## Additional files

### Supplementary files

• Transparent reporting form DOI: https://doi.org/10.7554/eLife.48434.019

### Data availability

All data generated or analysed during this study are included in the manuscript and supporting files.

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
