## [Decision Letter]

**Acceptance summary:**

This study has investigated the mechanism of action of a small molecule inhibitor of ricin toxicity called Retro-2. By comparing the genetic interaction profiles of Retro-2 to paired shRNA knockdowns, the authors find that Retro-2 action most resembles impairment of the TRC pathway for tail-anchored protein insertion. They use a focused mutagenesis strategy to identify and validate a resistance allele in ASNA1 (TRC40), further implicating the TRC pathway. Finally, reconstitution of the early steps of the TRC pathway shows that Retro-2 directly impairs substrate loading onto ASNA1/TRC40, while ASNA1/TRC40 containing the resistance mutation is not inhibited as effectively by Retro-2. Because trafficking pathways are impacted by TRC pathway impairment, and trafficking is a key requirement for Ricin toxicity, the authors provide strong evidence that Retro-2 acts by altering the levels of TA proteins (such as STX5) needed for ricin trafficking. The methods are of potential general use in identifying the mechanism and/or target of small molecules, and the biochemical data showing that the TRC pathway can be inhibited by Retro-2 is convincing. The study will be of wide interest to several fields.

**Decision letter after peer review:**

Thank you for submitting your article "Retro-2 prevents ricin toxicity by inhibiting ASNA1-mediated ER targeting and insertion of tail-anchored proteins" for consideration by *eLife*. Your article has been reviewed by three peer reviewers, one of whom is a member of our Board of Reviewing Editors, and the evaluation has been overseen by David Ron as the Senior Editor. The reviewers have opted to remain anonymous.

The reviewers have discussed the reviews with one another and the Reviewing Editor has drafted this decision to help you prepare a revised submission.

Summary:

This study has investigated the mechanism of action of the small molecule Retro-2 (and a related analog) in protecting cells from intoxication by ricin and shiga toxins by inhibiting retrograde trafficking from the early endosome to the ER (Stechmann et al., 2010). By comparing the genetic interaction profiles of Retro-2 to paired shRNA knockdowns, the authors find that the Retro-2 genetic profile resembles the profiles seen with impairment of the TRC pathway for tail-anchored protein insertion. They use a focused mutagenesis strategy to identify a resistance allele in ASNA1 (TRC40), further implicating the TRC pathway. Finally, reconstitution of the early steps of the TRC pathway shows that Retro-2 directly impairs substrate loading onto ASNA1/TRC40. Because trafficking pathways are indirectly impacted by TRC pathway impairment, and trafficking is a key requirement for Ricin toxicity, the authors suggest that Retro-2 acts by altering the levels of TA proteins (most notably STX5) needed for ricin trafficking. The methods are of potential general use in identifying the mechanism and/or target of small molecules, and the biochemical data showing that the TRC pathway can be inhibited by Retro-2 is well supported. However, the study does not yet make a sufficiently strong argument for the more important conclusion that TRC pathway inhibition (and particularly ASNA1/TRC40) is the decisive target by which Retro-2 protects from ricin toxicity. While the identification of Retro-2 as an inhibitor of the TRC pathway is a noteworthy advance, the study would need to provide stronger evidence for two of the study's key conclusions before publication in *eLife*: (i) that the effect on the TRC pathway is the primary basis for ricin protection; (ii) that ASNA1/TRC40 is likely to be the direct target. The following suggestions are offered to achieve this goal, in addition to a number of less central but potentially helpful suggestions for improvement.

Essential revisions:

1) Stechmann et al. previously showed that Retro-2 acts rapidly (within 30-60 minutes) to promote STX5 delocalization (and protection of cells from ricin), without inducing STX5 degradation. The speed of this effect and the absence of STX5 degradation would seem to be at odds with the proposed mechanism of action in the current study, which requires pre-existing STX5 to turn over concurrently with a biogenesis defect in STX5. Given that STX5 is a top hit in the chemical genetic screen, it seems that a major part of the effect of Retro-2 operates via STX5, so it is important to resolve the above discrepancy. The authors should test whether Retro-2 can delocalize a pre-existing pool of STX5, which might suggest an alternative (or additional) mechanism of action. Furthermore, the authors should directly determine whether endogenous STX5 is degraded upon treatment with Retro-2 (as predicted from their reporter assays) and whether this degradation occurs with a time course consistent with the observed time course of protection from ricin. These analyses are important in establishing a tighter correlation between the effect of Retro-2 on the TRC pathway and the effect on retrograde trafficking that is thought to underlie its protection from ricin. Finally, although it might not work for many reasons, the authors' case would be very strong if a Retro-2 resistant version of STX5 (e.g., by converting it into an ASNA1-independent substrate as in Figure 2B) could reverse the protective effect of Retro-2. This would effectively show that the key effect of Retro-2 is to inhibit ASNA1-dependent insertion of STX5.

2) A key part of the argument that the TRC pathway, and particularly ASNA1, is the functional target for Retro-2 is the discovery of a resistance allele of ASNA1. However, the cells with this resistance allele are not well characterized and the resistance phenotype is only cursorily examined. To rectify this problem, the authors should provide the following information: (i) whether the knock-in cells are clonal, whether they are heterozygous or homozygous with respect to the mutant allele, and whether resistance is observed in both cases; (ii) the steady-state expression level of A149V relative to WT ASNA1 in their cell lines; (iii) dose-response curves of Retro-2, DHQZ5, and DHQZ36.1 in assays of proliferation, ricin protection, and effect on the TA reporter(s) to more quantitatively characterize the extent to which the A149V ASNA1 mutation confers resistance. The concern here is that the resistance seen with A149V is due to a relatively trivial consequence of this mutant protein being mostly non-functional.

3) Related to the concern raised in point 2, the authors need to better characterize how the A149V mutation confers resistance to Retro-2. Ideally, the authors would measure the binding affinity of Retro-2, DHQZ5, and DHQZ36.1 for WT and A149V ASNA1. In the absence of a direct binding assay (which is acknowledged to be challenging and possibly not feasible in the context of this study), the authors should test whether A149V ASNA1 is resistant to DHQZ36.1 in the biochemical assay used in Figure 5D, E. In particular, it is critical to determine whether A149V ASNA1 is functional with respect to substrate binding, whether it promotes TA insertion, and whether its capacity to mediate ATP hydrolysis is retained. Demonstration that A149V mutant ASNA1 is functional but resistant to the effects of Retro-2 would help to strengthen the authors' case that this mutation is a true resistance allele.

---

## [Author Response]

Essential revisions:1) Stechmann et al. previously showed that Retro-2 acts rapidly (within 30-60 minutes) to promote STX5 delocalization (and protection of cells from ricin), without inducing STX5 degradation. The speed of this effect and the absence of STX5 degradation would seem to be at odds with the proposed mechanism of action in the current study, which requires pre-existing STX5 to turn over concurrently with a biogenesis defect in STX5. Given that STX5 is a top hit in the chemical genetic screen, it seems that a major part of the effect of Retro-2 operates via STX5, so it is important to resolve the above discrepancy. The authors should test whether Retro-2 can delocalize a pre-existing pool of STX5, which might suggest an alternative (or additional) mechanism of action. Furthermore, the authors should directly determine whether endogenous STX5 is degraded upon treatment with Retro-2 (as predicted from their reporter assays) and whether this degradation occurs with a time course consistent with the observed time course of protection from ricin. These analyses are important in establishing a tighter correlation between the effect of Retro-2 on the TRC pathway and the effect on retrograde trafficking that is thought to underlie its protection from ricin. Finally, although it might not work for many reasons, the authors' case would be very strong if a Retro-2 resistant version of STX5 (e.g., by converting it into an ASNA1-independent substrate as in Figure 2B) could reverse the protective effect of Retro-2. This would effectively show that the key effect of Retro-2 is to inhibit ASNA1-dependent insertion of STX5.

We thank the reviewers for highlighting this previous result.

To directly address the question of whether there are additional (ASNA1-independent) mechanisms by which Retro-2 might act that contribute to its effect on ricin toxicity, we used CRISPR-mediated HDR combined with single-cell cloning to create homozygous resistant ASNA1 mutant (A149V) K562 cells (Figure 4—figure supplement 2A). We find that at relevant timepoints and doses, we observe no significant protective activity of Retro-2 in the presence of the A149V allele (Figure 4—figure supplement 2D). We think this is the most convincing demonstration that Retro-2 activity goes through ASNA1. While we cannot definitively comment on the acute effects of Retro-2 observed by Stechmann et al. within 30 minutes, we think the fact that our ASNA1 mutant prevents the growth effects and ricin protection of Retro-2 suggests that either (1) the acute phenotypes previously observed should also be dependent on ASNA1 activity (for example, through an acute stress response) or (2) if there is an additional STX5-dependent, ASNA1-independent activity, it is likely not relevant to for ricin protection.

We also note that since we observe effects of the Retro-2 analog DHQZ36.1 in a defined in vitro system using SEC61B as a substrate in the absence of STX5 (Figure 5—figure supplement 1F), we have established that there is a STX5-independent activity of Retro-2 analogs. In sum, we believe that Retro-2 is very unlikely to act directly on STX5 (i.e. outside of its time as a TRC pathway client) as part of its ricin protection mechanism.

The reviewers also request to see the effects of Retro-2 on endogenous STX5 to complement our reporter assays. As noted, the original publication of Retro-2 reported no changes in total STX5 levels upon Retro-2 treatment, and our preliminary results also indicate no changes as measured by Western blot to total STX5 upon Retro-2 treatment or knockdown of ASNA1 in the timeframes examined here (data not included). Our microscopy staining for endogenous STX5 (Figure 3), in contrast, shows a dramatic loss of endogenous STX5 staining with either Retro-2 treatment or ASNA1 knockdown.

2) A key part of the argument that the TRC pathway, and particularly ASNA1, is the functional target for Retro-2 is the discovery of a resistance allele of ASNA1. However, the cells with this resistance allele are not well characterized and the resistance phenotype is only cursorily examined. To rectify this problem, the authors should provide the following information: (i) whether the knock-in cells are clonal, whether they are heterozygous or homozygous with respect to the mutant allele, and whether resistance is observed in both cases; (ii) the steady-state expression level of A149V relative to WT ASNA1 in their cell lines; (iii) dose-response curves of Retro-2, DHQZ5, and DHQZ36.1 in assays of proliferation, ricin protection, and effect on the TA reporter(s) to more quantitatively characterize the extent to which the A149V ASNA1 mutation confers resistance. The concern here is that the resistance seen with A149V is due to a relatively trivial consequence of this mutant protein being mostly non-functional.

We thank the reviewers for raising these concerns about the resistance allele of ASNA1. We now present additional experiments both in cells and in the reconstituted system to further establish that the A149V allele remains functional.

Our original manuscript demonstrated the activity of the A149V mutant in a mixed population (Figure 4D), which contained wildtype, A149V, and knockout copies of ASNA1 (Figure 4—figure supplement 1B). As the ASNA1 locus in K562 cells is triploid, this pool contained many different genotypes. Our results indicated that the pool as a whole was resistant to Retro-2 activity, which could be attributed to any of these genotypes.

Thus, to directly address the reviewer’s question, we isolated single-cell clones and selected homozygous mutant lines with defined genotypes (Figure 4—figure supplement 2A). When compared to a wildtype ASNA1 clonal line (created using a safe-targeting guide) and a synonymous, homozygous A149A ASNA1 clonal line, we observe that the homozygous A149V ASNA1 clonal line does not affect protein abundance (Figure 4—figure supplement 2B) but is highly resistant to both the protective activity of Retro-2 and the toxicity of DHQZ36.1 (Figure 4—figure supplement 2C, D). Note that we also observe no growth phenotype or resistance to ricin in the absence of Retro-2 analogs, suggesting that the A149V allele can fully complement these phenotypes.

Additionally, we transduced these clonal cell lines with the GFP-2A-RFP-SEC61B_TMD_ reporter. This more sensitive measure of ASNA1 function revealed that ASNA1 A149V caused a partial loss of function under basal conditions but still conferred about a ten-fold increase in EC_50_ for Retro-2 and DHQZ36.1 (Figure 4E; Figure 4—figure supplement 2E, F).

3) Related to the concern raised in point 2, the authors need to better characterize how the A149V mutation confers resistance to Retro-2. Ideally, the authors would measure the binding affinity of Retro-2, DHQZ5, and DHQZ36.1 for WT and A149V ASNA1. In the absence of a direct binding assay (which is acknowledged to be challenging and possibly not feasible in the context of this study), the authors should test whether A149V ASNA1 is resistant to DHQZ36.1 in the biochemical assay used in Figure 5D, E. In particular, it is critical to determine whether A149V ASNA1 is functional with respect to substrate binding, whether it promotes TA insertion, and whether its capacity to mediate ATP hydrolysis is retained. Demonstration that A149V mutant ASNA1 is functional but resistant to the effects of Retro-2 would help to strengthen the authors' case that this mutation is a true resistance allele.

We appreciate the reviewer’s suggestions. We agree that direct binding assays of Retro-2 to ASNA1 are beyond the scope of the current study, but as suggested, we pursued the functional experiments in our purified biochemical system.

We first created and purified the homologous ASNA1 mutation (A143V) in zebrafish protein (Figure 5—figure supplement 2A). In our biochemical experiments, mutant ASNA1 was able to accept substrate transfer from SGTA, demonstrating that the mutant remains functional in this reconstituted reaction (Figure 5F). While the presence of DHQZ36.1 in this experiment inhibits substrate transfer of the wildtype ASNA1, we observed a reduced inhibition for the mutant ASNA1 (Figure 5F).

These results in a reconstituted pathway with five components demonstrate conclusively that the A149V ASNA1 mutant is functional and resistant to DHQZ36.1.